# Milling Al520-MMC Reinforced with SiC Particles and Additive Elements Bi and Sn

**DOI:** 10.3390/ma15041533

**Published:** 2022-02-18

**Authors:** Mahmoud Alipour Sougavabar, Seyed Ali Niknam, Behnam Davoodi, Victor Songmene

**Affiliations:** 1School of Mechanical Engineering, Iran University of Science and Technology, Tehran 13114-16846, Iran; mahmoudalipour1990@gmail.com (M.A.S.); bdavoodi@iust.ac.ir (B.D.); 2Mechanical Engineering Department, Ecole de Technologie Superieure (ETS), 1100 Notre-Dame St W, Montreal, QC H3C 1K3, Canada; victor.songmene@etsmtl.ca

**Keywords:** metal matrix composite, aluminum, reinforcing particles, machinability, tool flank wear, surface quality

## Abstract

In recent years and due to advanced fabrication techniques of composites, many of these functional materials have been brought to the forefront with more benefits. Amongst composites, special attention has been paid to metal matrix composites (MMCs). Reinforced aluminum MMCs with nanoparticles are among the new MMCs with a wide range of industry applications. The combination of aluminum as a soft, lightweight, and low-strength material with silicon carbide (SiC), bismuth (Bi), and tin (Sn) particles, which are hard and high-strength materials, may lead to the generation of high-strength and lightweight material, which can be classified as difficult to cut material. According to literature, limited studies have been reported on the effects of various reinforcing elements on the machinability of Al-MMC, in principle tool wear morphology and size and surface quality. According to statistical analysis, the effect of cutting parameters and reinforcing particles on the surface quality attributes is not statistically significant. In contrast, the effect of cutting parameters and reinforcing particles on the tool flank wear is significant and reliable. In addition, it is observed that the reinforcing particles and cutting speed have the most significant effects, and the lubrication mode has a minor impact on the tool flank wear.

## 1. Introduction

In recent years, the automobile and aerospace industries have used particulate reinforced aluminum metal matrix composites (MMCs) because of their low specific weight, high wear resistance, low thermal expansion, and improved mechanical properties [1,2,3]. Industrial interests have led to in-depth research into the machining of composite-based metals [4,5,6]. However, the main observations are that reinforcing the base metal with hard particles may complicate machining operations, while causing higher flank wear, poor surface quality, and higher manufacturing costs. Most of the reported studies in the literature [5,6,7,8] state that carbide and polycrystalline diamond (PCD) tools are mainly used in machining MMCs. The majority of notable works are on the turning operation. Many studies claim that carbide tools are suitable alternatives for CBN and PCD tools under certain conditions [4,7,8]. An essential step towards maximizing efficiency and stability when machining metal-based composites is to optimize the machining parameters. Therefore, a suitable selection of machining parameters is crucial to create an economical machining process. Generally, limited studies have been reported in the area of the milling process [8,9,10,11]. It has been observed that deteriorated surface quality may occur when reinforcing fibers or particles are removed during milling operations [5,12,13]. In principle, an adequate selection of cutting conditions has a significant role in improving machining performance, flank wear control, and reduction. Many studies have reported the machining of superalloys and MMCs using ceramic tools [12,14]. According to literature, the initial wear of carbide tools at intermediate to high cutting speeds has not been studied in a broad scope [11,15]. Most of the reported research studies on the machinability of aluminum substrate composite have revealed the significant effects of depth of cut and feed rate. Several studies on the turning process show that increasing the depth of cut would increase the cutting force. It was stated that the optimum surface quality occurs at the lowest level of depth of cut. This underlines that depth of cut has a significant effect on cutting force [11,15]. In addition, limited studies have been reported on the effects of various reinforcing elements on the machinability of Al-MMC, in principle flank wear morphology and size. Furthermore, the fabrication process of such materials with additive particles is still the subject of additional studies [16]. For example, Keremer et al. analyzed the machinability of aluminum matrix composite with added SiC. Elevated wear was observed when the reinforcing elements exceeded a critical limit in MMC [17]. According to K. Qiu’s study, A356 with 0.5% Sn positively influenced high-temperature properties and improved flow behavior and mold-filling capabilities [18]. Moreover, bismuth (Bi) was used as an alloying element in wrought aluminum alloys to improve chip breakage and lubrication [19]. Bi was also added to Al alloys to prevent embrittlement by sodium [20] and to disrupt the formation of oxide defects in Al alloys [21].

Additionally, those work parts with Bi explored the lowest cutting force and better surface roughness in Al-11Si-2Cu cast alloy as compared to those with Sb and Sr [22]. In this regard, in this study, Al520 was used as the base material, and in combination with reinforcing elements (10% SiC, 1% Bi, 1% Sn), three types of A520-MMC were fabricated based on the addition of various reinforcing elements. Various cutting parameters were used, and the experimental plan was conducted in Taguchi L16 for each tested material. Their machinability was compared with Al520 as the pure material with no added particulates. The machinability evaluation was performed using the surface roughness attributes (Ra), cutting forces, tool wear morphology, size, and dust emission. However, only recorded values of surface roughness and flank wear will be presented in this study, and other machinability attributes will be studied in future articles. In the next section, experimental work is presented. Finally, Section 3 is devoted to results and discussion, followed by the conclusion.

## 2. Materials and Methods

The blocks of composite materials tested (Table 1) were fabricated with a casting system depicted in Figure 1. The Al520 was used as the base material to produce the composite blocks tested. The SiC particles with an average diameter of 40 μm were used in several blocks. The SiC elements underwent the heat treatment process at 1000 °C for 120 min to break the bonds and prevent colliding, sticking, and abrasion. The SiC particles were then preheated at 400 °C for 60 min. Besides the pure block of Al520, the second block was reinforced with 10% SiC, while the third and fourth ones were fabricated and reinforced with Bi and Sn and 10% SiC (Table 1). The dimensions of the tested work parts are shown in Figure 1. At the first step of the fabrication process of MMC blocks, the base material Al520 alloy was placed in the containers at the desired weight percentage. After melting the alloy, 50 g of SiC and 5 g of Bi and Sn were gradually added to each mixture at 750 °C. After injecting SiC for 15 min, the mixer was operated at 680 rpm. The SF6 shielding gas was injected into the container to prevent the melt and oxide formation reaction during the entire fabrication process. The length of the handle was 28.6 cm and it had four blades, and the width of the blades was 1 and 2 cm, respectively, and the blades had an angle of 30 degrees concerning the direction of the stirrer handle. The mixer was coated with a mixture of zircon liquid to prevent it from mixing with the melt. Before pouring the melt into the mold, the mixture was preheated to avoid sticking between the melt and the mold. The melting mixture was poured into the mold at the end of the stirring process.

Upon completing Al520 + 10% SiC casting process, a section of the work part was separated as a sample to ensure the correct distribution of particles and quality production of the fabricated work part. It was polished and prepared for characterization using SEM. As a result, it can be seen that except for a few points where these particles are stuck together and prevention is almost impossible, the uniform homogeneity was observed through the work part. The quantometry examination of the tested part was conducted at the temperature: 25 °C and the humidity: 41%. Table 2 presents the results of quantometry.

The cutting parameters used in this work are presented in Table 3. The experimental parameters were selected according to industrial recommendations by the cutting tool manufacturer. The MeC Green biofluid was used as the lubricant in the wet cutting conditions. To construct an excellent experimental plan with fewer experiments needed, the L16 Taguchi orthogonal array was used for each material (Table 4). The experiments were repeated twice. The statistical analysis was later performed in commercial software Minitab and Stratigraphic.

The cutting operations were performed on a 3-axis CNC machine tool (power: 50 kW, speed: 28,000 rpm; torque: 50 Nm) under different lubrication conditions, including dry and wet (Figure 2). Iscar cutting tool: HP E90AN-D.50-2-C.50-7C CUTTER and Iscar insert HM90 APKT 1003 PDR; IC 908 were used in cutting tests. A new and sharp insert was used in each test. The characteristics of the inserts used are presented in Table 5.

After completing the milling tests, the flank wear rate and morphology and the Ra readings were evaluated under different cutting conditions. Abrasion is the dominant wear mechanism due to the interaction between SiC particles in the work part and the insert surface. According to Manna et al., the leading cause of flank wear in machining MMCs is the collision of SiC particles with the surface of the carbide tool [23]. Therefore, the surface roughness measurement was conducted on the profilometer Mitutoyo SJ 400. To better elaborate the effect of cutting parameters and flank wear mode on the surface quality, the surface roughness measurement was conducted within the sides (Enter, Middle, Exit) of the work part in each sample, and individual sets of analysis were performed.

## 3. Results and Discussion

The milling tests were performed under the cutting conditions presented in Table 3. In addition, the surface quality attributes (Ra, Rz) and the tool flank wear morphology and size were recorded under each condition. This includes accurate characterization of tool flank wear morphology and size in the inserts using SEM and EDS.

### 3.1. Surface Quality Attributes (Ra, Rz)

According to the tested regions (Figure 3), the surface roughness measurements were conducted in three sections (Enter, Middle, Exit) in all tested materials. In addition, a separate analysis corresponding to each tested material is presented in the following sections.

#### 3.1.1. Surface Quality Attributes (Ra, Rz) in Al520

Based on Figure 4, it is observed that the effects of cutting parameters on Ra and Rz are nonlinear. Therefore, it can be stated that since a sharp cutting edge was used, the recorded values of Ra are less than the measurements made in the Exit and Middle sections. In addition, according to the correlation of determination presented in Figure 4a, the R^2^ = 54.98% states that the effect of cutting parameters on Ra in Al520 is not statistically significant. In other words, the effects of cutting parameters such as cutting speed, feed rate, and depth of cut are insignificant. Furthermore, as shown in Figure 4b, the effect of cutting parameters on Rz in Al520 is also not statistically significant.

The 2D contour plots of Rz and Ra against cutting speed and feed rate per tooth are shown in Figure 5. It is shown that no mathematical formulation can be formulated between both responses and cutting speed and feed per tooth. This phenomenon could be due to interaction effects between cutting parameters. However, as expected, the worst surface quality was observed at the highest levels of cutting speed and feed per tooth when machining Al520.

#### 3.1.2. Surface Quality Attributes (Ra, Rz) in Al520 + 10% SiC

Based on the observations made in Figure 6, it can be stated that cutting parameters have nonlinear effects on Ra and Rz values when machining Al520 + 10% SiC. In other words, a linear fluctuation can be observed under various levels of cutting parameters. Based on ANOVA results, both Ra and Rz are not statistically sensitive to variation of cutting parameters. One reason for this phenomenon could be due to this material’s composition and mechanical behavior. The presence of SiC as a reinforcing material in the material matrix may tend to generate variation in the surface topology and roughness profile. Since the SiC dispersion is nonhomogeneous in the matrix structure, the collision between the SiC and the cutting tool cannot be predicted. As a result, high wear and mechanical damage to the tool’s cutting edge may occur. In other words, it becomes somehow impossible to analyze and express the scientific and systematic overview of the effects of cutting parameters on the surface profile and surface quality attributes.

Similar to Figure 5, it can be stated that no mathematical formulation can be formulated between both surface roughness responses and cutting parameters (Figure 7). This phenomenon could be related to the interaction between the cutting parameters and the presence of SiC as a reinforcing element in the material matrix. Furthermore, since the SiC dispersion is nonhomogeneous in the matrix structure, the collision between the SiC and the cutting tool cannot be predicted. As a result, high wear and mechanical damage to the tool’s cutting edge may occur, leading to a nonlinear state due to the presence of SiC and interaction effects of cutting parameters when machining Al520 + 10%SiC.

#### 3.1.3. Surface Quality Attributes (Ra, Rz) in Al520 + 10%SiC + 1%Bi

According to Figure 8, similar observations were made compared to previous materials. According to statistical analysis and analysis of variance, it can be declared that the effects of cutting parameters on Ra and Rz in Al520 + 10%SiC + 1% Bi are not statistically significant. In other words, the effects of cutting parameters such as cutting speed, feed rate, and depth of cut are insensitive. Therefore, the addition of Bi to the composition of the MMC has not led to the optimization of the statistical modeling process.

Similar to Figure 5 and Figure 7, it can be stated that no mathematical formulation can be formulated between both surface roughness responses and cutting parameters (Figure 9) due to interaction between the cutting parameters and the presence of SiC and Bi as reinforcing elements in the material matrix. Furthermore, since all three elements SiC, Sn, and Bi disperse nonhomogeneously in the matrix structure, the collision between the Si and Bi and the cutting tool cannot be predicted when machining Al520 + 10%SiC + 1%Bi. In addition, contrary to Figure 5 and Figure 7, the highest values of Ra and Rz have not resulted in the highest levels of feed per tooth and cutting speed. However, lower roughness was recorded when machining Al520 + 10%SiC + 1%Bi compared to readings made when machining other tested materials.

#### 3.1.4. Surface Quality Attributes (Ra, Rz) in Al520 + 10%SiC + 1%Sn

Based on Figure 10, similar to other materials tested, Ra and Rz are not statistically sensitive to variation of cutting parameters. In other words, cutting parameters have nonlinear effects on the tested part. Again, similar to other materials, this phenomenon can be considered the result of the initial friction of the tool colliding with the work part.

Irrespective of the material used, it can be stated that no mathematical formulation can be formulated between both surface roughness responses and cutting parameters (Figure 11) due to interaction between the cutting parameters and the presence of SiC and Sn as reinforcing elements in the material matrix. Furthermore, since SiC and Sn disperse nonhomogeneously in the matrix structure, the collision between Si and Sn and the cutting tool cannot be predicted when machining Al520 + 10%SiC + 1%Sn. In addition, contrary to Figure 6 and Figure 8, the highest values of Ra and Rz were not recorded in the highest levels of feed per tooth and cutting speed.

Finally, it can be stated that despite the material tested, the effects of reinforcing particles on both Ra and Rz are not statistically significant. Knowing that fabricated MMCs have heterogeneous distribution inside the material, the impact of the tooltip during machining such materials causes mechanical damage, abrasion, and high surface roughness. Furthermore, except for minor cases, in general, due to the collision of SiC with the tip of the cutting tool, rapid tool wear or breakage is expected. Consequently, more extensive chip formation, friction, and forces are expected. As a result, deteriorated surface quality is expected. One solution to overcome this problem is to use appropriate tool path strategies, adequate cutting parameters, suitable cutting tools, and tool geometry.

Furthermore, additional tests are still needed to confirm these statements due to uncontrolled and unpredicted collision between the tooltip and reinforcing particles which may tend to hasten tool wear and, in some cases, tool breakage. Moreover, according to the statistical studies performed on each material and examining the effect of different materials and cutting parameters on the surface quality, it can be stated that the effect of cutting parameters on Ra and Rz is statistically insensitive in all studied conditions and is uncontrollable (Table 6). Therefore, no numerical modeling can be formulated between Ra and Rz and the cutting parameters used. Thus, although the addition of Bi and Sn in both fabricated parts improved surface quality, no clear statistical justification can be made concerning the effects of cutting parameters on the surface quality attributes.

### 3.2. Tool Flank Wear (Measurement)

#### 3.2.1. Tool Flank Wear (Measurement) in Al520

To determine the cutting tool life and observe the overview of the tool flank wear profile, the amount of tool flank wear should be monitored and measured. For this purpose, the flank side of the cutting tool was examined when various levels of cutting parameters were used. As can be seen in Figure 12a, a high correlation of determination (R^2^ = 98.52%) states that the variation of cutting parameters can control the tool flank wear when machining Al520. The cutting speed and lubrication have the most effects on the tool flank wear. In addition, higher tool flank wear was observed under dry mode. Thus, in general, it can be stated that a mathematical relationship can be formulated between tool flank wear and cutting parameters. However, based on the contour plot of tool flank wear in Figure 12b, there is no definite and pragmatic relationship between the variation of cutting speed and feed per tooth and the resulting values of tool flank wear when machining Al520. However, it can be understood that the higher cutting speed may tend to elevate tool flank wear. In fact, except for minor cases, the highest values of tool flank wear were observed between cutting speed 150–180 m/min and feed per tooth 0.08–0.10 mm, which is in agreement with Figure 12a.

#### 3.2.2. Tool Flank Wear (Measurement) in Al520 + 10% SiC

According to Figure 13a, although the presence of SiC particles increases the tool flank wear as compared to the readings made in Figure 12a, tool flank wear can still be formulated as a function of cutting parameters owing to the high value of the correlation of determination (R^2^ = 94.77%). Furthermore, it is well known that a higher level of cutting speed may tend to increase tool flank wear. However, based on the tool flank wear chart in Figure 13b, it can be seen that cutting parameters have nonlinear effects on tool flank wear values in the machining of Al520 + 10%SiC. As expected, the highest values of tool flank wear were recorded when Al520 + 10% SiC was compared to the reading made when machining other materials. This phenomenon could be attributed to the presence of SiC in the work part matrix, which tends to degrade the tool life when collisions occur between the cutting tool and the work part. Therefore, it can be inferred that SiC in the matrix structure has more effects than lubrication mode on the tool flank wear. Furthermore, the highest values of tool flank wear were recorded at cutting speed 120/m/min and feed per tooth range 0.10–0.11 mm, which can also be confirmed with Figure 13a.

#### 3.2.3. Tool Flank Wear (Measurement) in Al520 + 10%SiC + 1%Bi

Based on the tool flank wear chart in Figure 14a and statistical analysis, the tangible statistical effects of cutting parameters on the tool flank wear can be expressed when machining Al520 + SiC 10% + Bi 1%. It can be concluded that adding Bi to the material’s structure reduces the tool flank wear range between 45 and 80 microns. As a result, this element can reduce wear and increase the surface quality of this aluminum-based composite with SiC-reinforcing particles. According to Figure 14b, it can be seen that cutting parameters have nonlinear effects on tool flank wear values in the machining of Al520 + 10%SiC + 1%Bi. Furthermore, the presence of the Bi element in this composite led to minor mechanical damage in the inserts. Therefore, it can be considered that the quality of surfaces in this material has also been improved by reducing tool flank wear.

#### 3.2.4. Tool Flank Wear (Measurement) in Al520 + 10%SiC + 1%Sn

Based on the tool flank wear chart in Figure 15a and statistical analysis, the noticeable statistical effect of cutting parameters on the tool flank wear changes in Al520 + SiC 10% + Sn 1% can be observed. According to Figure 15a, it can be concluded that adding Sn to this material reduces the tool flank wear range between 40 and 65 microns; among all of these materials that we added (SiC, Bi, Sn), the lowest amount of tool flank wear was observed after adding Sn. Whenever the machining changes from dry to wet, the tool flank wear rate decreases, and as a result, the tool life increases. According to Figure 15b, it can be seen that cutting parameters have nonlinear effects on tool flank wear values in the machining of Al520 + 10%SiC + 1%Bi. The amount of tool flank wear due to the Sn element in this composite material has decreased compared to composite material without Sn. Therefore, it can be concluded that adding the Sn element to this composite material reduces the amount of mechanical damage to the tool. As a result, the tool life can be increased. The highest tool flank wear was observed at a 120 m/min cutting speed.

Due to the heterogeneous distribution of SiC particles and the sudden collision of the cutting tool with SiC particles in the tested materials, it cannot be stated that a higher cutting speed may lead to higher tool flank wear size. Regardless of the type of tested material, it can be seen that more tool flank wear was observed in dry mode. When the depth of cut increases, the tool flank wear was also raised at some levels, but interaction effects between the depth of cut can be observed. The importance of each cutting parameter on the tool flank wear when machining the tested materials is presented in Table 7. The ascending order of numbers refers to most minor effective parameters on the tool flank wear control. It can be seen that except Al520, the cutting speed is the governing factor on the variation of flank wear in all other reinforced materials. The correlation of determination (R^2^) associated with each material indicates that the tool flank wear in all materials can be formulated as a function of cutting parameters tested. However, the order of importance of each cutting parameter on the tool flank wear is not identical in each material.

As noted earlier, the characterization of tool flank wear morphology is beyond the scope of this work. However, based on experimental observations in dry and lubricated milling under various cutting speeds, adhesion and abrasion were found in almost all cutting conditions. It is believed that abrasion initially occurred, whereas coating removal also occurred. This is followed by adhesion, but the experimental results show higher wear with hard additive particles (10%SiC, 1%Bi, 1%Sn). This phenomenon increases mechanical pressure and surface structure changes at high temperatures. The experimental result denoted that fabricating the Al520-MMC with such hard particles tends to generate a high-hardness and high-strength MMC, which may increase the tool wear compared to measurements made when machining Al510. However, the use of Sn and Bi tends to lower values of tool flank wear in the tested parts.

## 4. Conclusions

This study aims to present the machinability of Al520-MMC, reinforced with Sic, Bi, and Sn as reinforcing elements. Besides the fabrication of Al-MMC blocks, the effects of cutting parameters and reinforcing elements on the surface quality elements (Ra and Rz) and tool flank wear were investigated in this work. According to statistical analysis, the effect of cutting parameters and reinforcing particles on the surface quality attributes (Ra and Rz) is not statistically significant. In contrast, the effect of cutting parameters and reinforcing particles on the tool flank wear is significant and reliable. In addition, it is observed that the reinforcing particles and cutting speed had the most significant effects, and the lubrication mode had minor effects on the tool flank wear. However, no clear relationship could be formulated between recorded values of tool flank wear when machining base material (Al520) and reinforced Al520-MMCs with Sic, Bi, and Sn. Other observations and conclusions can be drawn as follows:The effects of cutting parameters on Ra and Rz were statistically insignificant, while the R2 was almost below 60% in all cases. Thus, it can be stated that despite the reinforcing elements used, both surface roughness attributes cannot be controlled by the cutting parameters used.It was observed that the wear increases at a higher cutting speed irrespective of the cooling method used.In some cases and due to the presence of SiC in the tested work parts and its heterogeneous distribution in the base material, inevitable collisions may occur between the cutting tooltip and the reinforcing elements. This phenomenon may tend to increase the wear drastically.The use of Bi and Sn in the matrix structure of the work part led to better surface quality than that observed when machining Al520.It was found that despite the material used, tool flank wear can be statistically controlled by cutting parameters. The effects of cutting speed and lubricant on the tool flank wear are very tangible. Depth of cut had the most negligible effects compared to other cutting parameters.Compared with the recorded tool flank wear in machining Al520 + 10% SiC, the use of Bi and Sn in the matrix structure of the work part led to better surface quality, and tool flank wear was decreased by around 50%.Lower tool flank wear was observed under wet machining than was observed under dry mode despite the reinforcing elements used.It can be stated that no clear relationship could be established between recorded values of tool flank wear in the base material (Al520) and reinforced composites with SiC, Bi, and Sn particles. However, this phenomenon could be related to different effects of each cutting parameter on the recorded values of tool flank wear when machining Al520-MMCs, reinforced with various elements. This observation agrees with the findings reported in Table 7.

## Figures and Tables

**Figure 1 materials-15-01533-f001:**
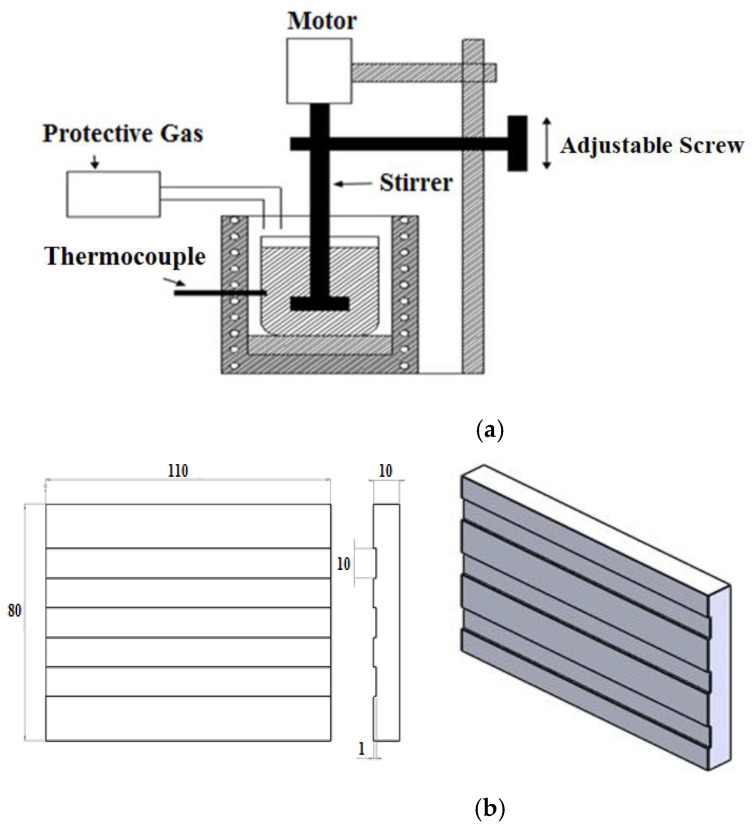
(**a**) Schematic of the casting system used; (**b**) the dimensions of the tested work parts (unit is mm).

**Figure 2 materials-15-01533-f002:**
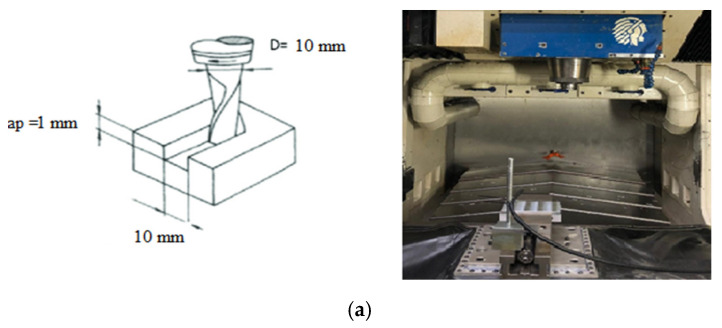
Schematic and practical overview of machining set up: (**a**) dry machining; (**b**) wet machining.

**Figure 3 materials-15-01533-f003:**
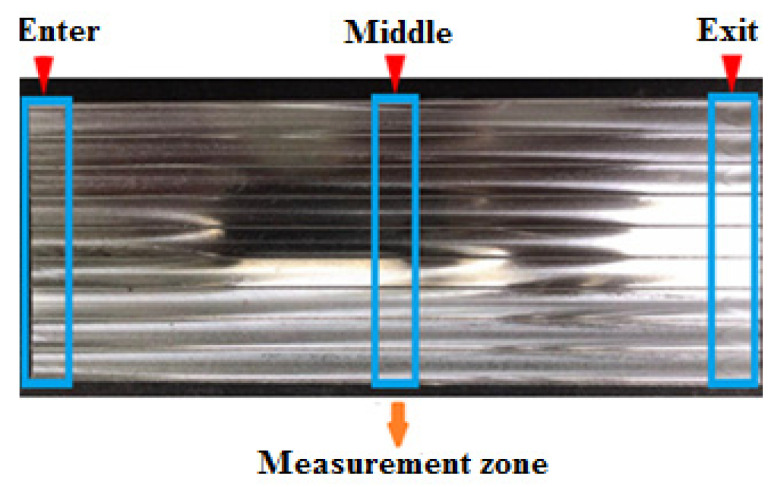
Surface roughness measurement zones.

**Figure 4 materials-15-01533-f004:**
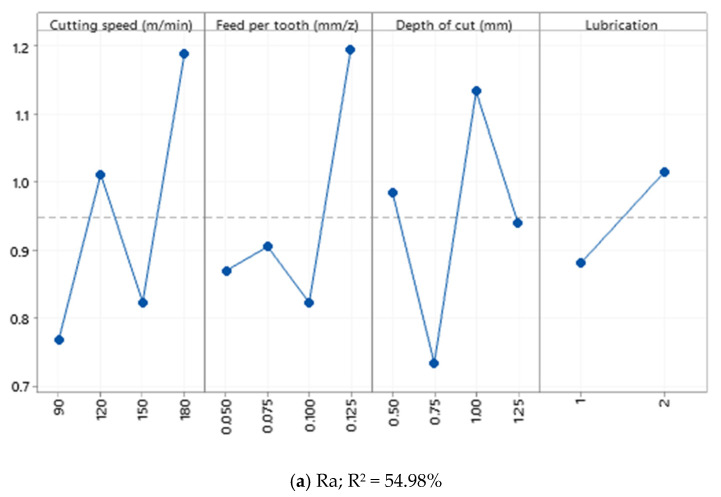
Main effect plot of (**a**) Ra; (**b**) Rz recorded from the Enter section.

**Figure 5 materials-15-01533-f005:**
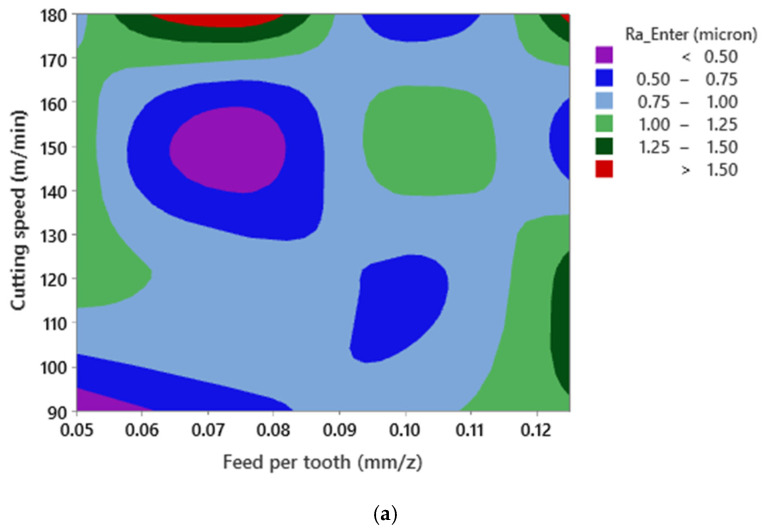
Surface plot of Ra—Enter versus cutting speed (m/min), and feed per tooth (mm), for (**a**) Ra—Enter and (**b**) Rz—Enter in Al520.

**Figure 6 materials-15-01533-f006:**
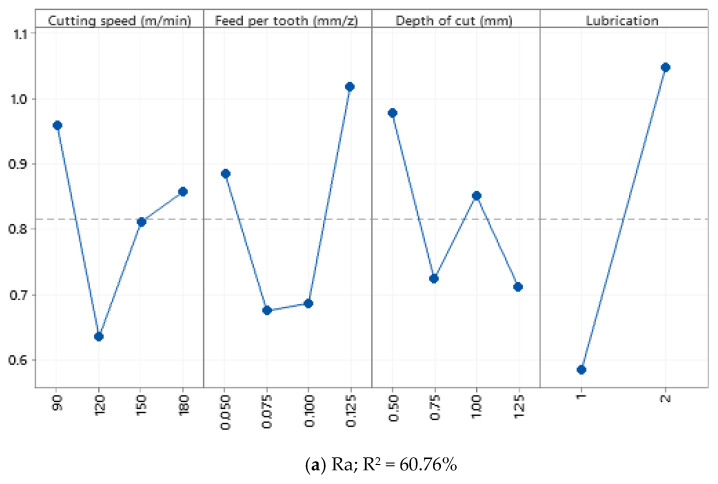
Main effect plot of (**a**) Ra and (**b**) Rz recorded from the Enter section.

**Figure 7 materials-15-01533-f007:**
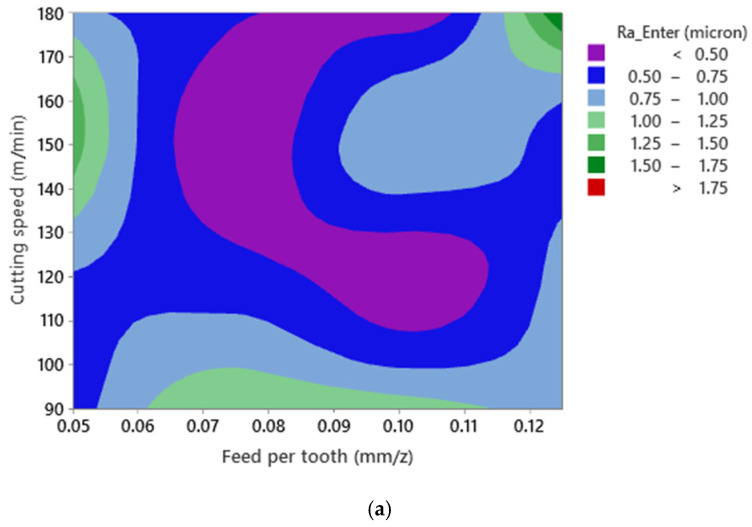
Surface plot of Ra—Enter versus cutting speed and feed per tooth for (**a**) Ra—Enter and (**b**) Rz—Enter in Al520 + 10% SiC.

**Figure 8 materials-15-01533-f008:**
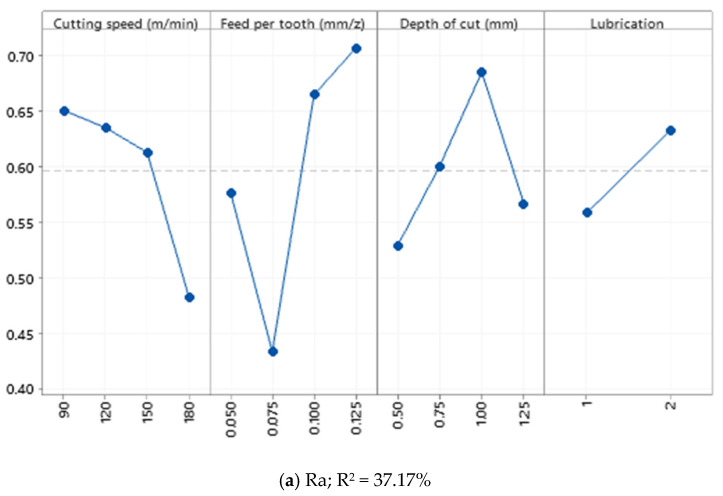
Main effect plot of (**a**) Ra; (**b**) Rz recorded from the Enter section.

**Figure 9 materials-15-01533-f009:**
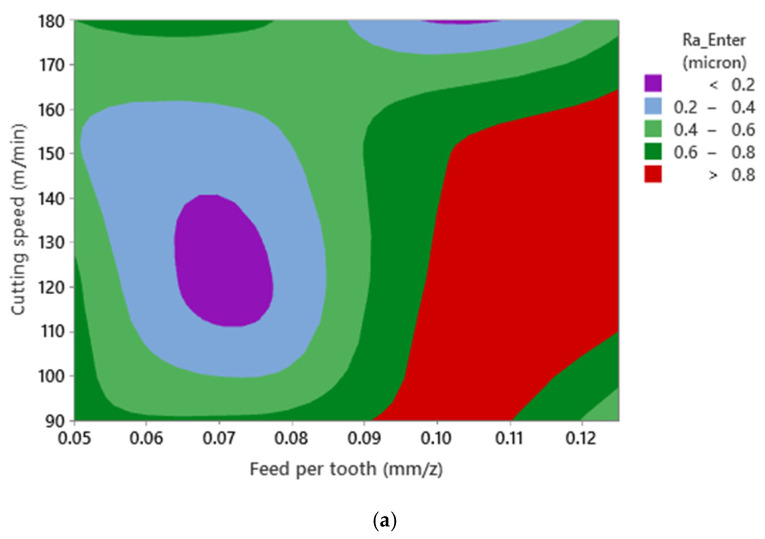
Surface plot of Ra—Enter versus cutting speed and feed per tooth for (**a**) Ra—Enter and (**b**) Rz—Enter in Al520+ 10%SiC + 1%Bi.

**Figure 10 materials-15-01533-f010:**
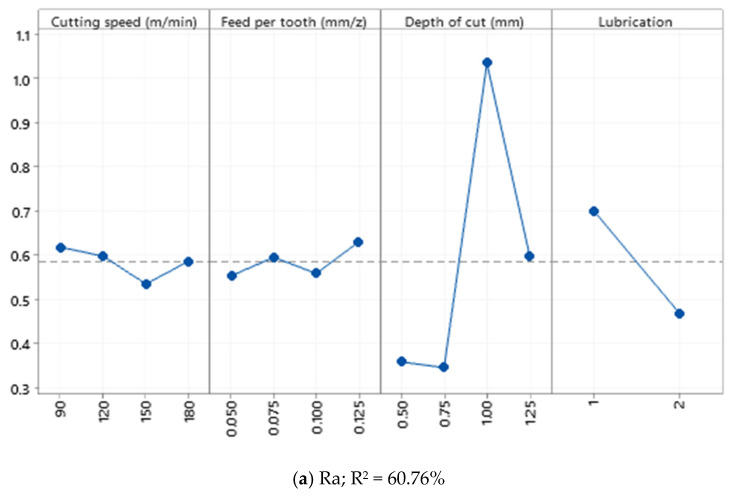
Main effect plot in the Enter section; (**a**) Ra; (**b**) Rz.

**Figure 11 materials-15-01533-f011:**
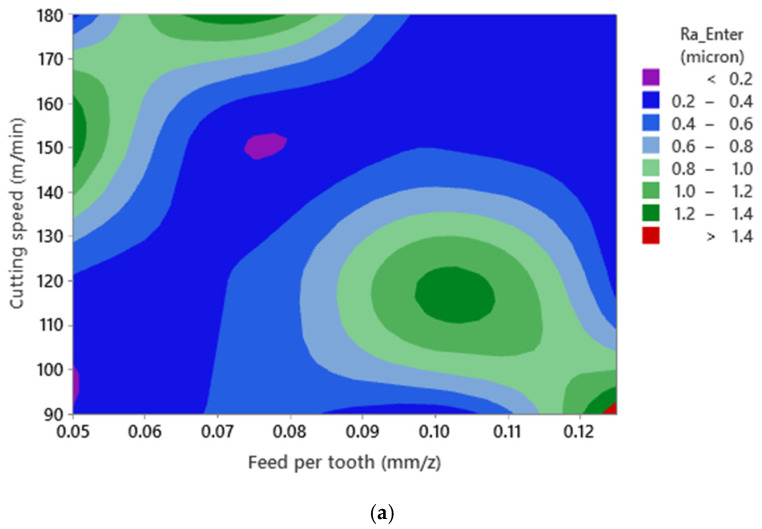
Surface plot of Ra—Enter versus cutting speed and feed per tooth for (**a**) Ra—Enter and (**b**) Rz—Enter in Al520 + 10%SiC + 1%Sn.

**Figure 12 materials-15-01533-f012:**
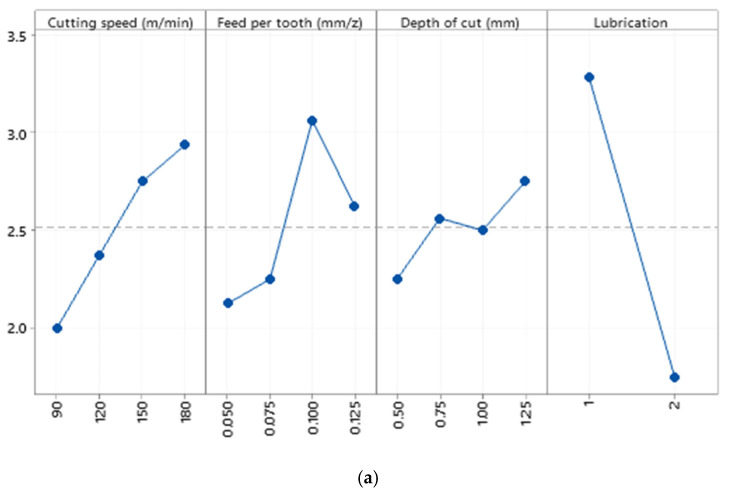
(**a**) The main effects plot of tool flank wear when machining Al520 + 10% SiC (R^2^ = 98.52%), (**b**) surface plot of tool flank wear versus cutting speed and feed per tooth.

**Figure 13 materials-15-01533-f013:**
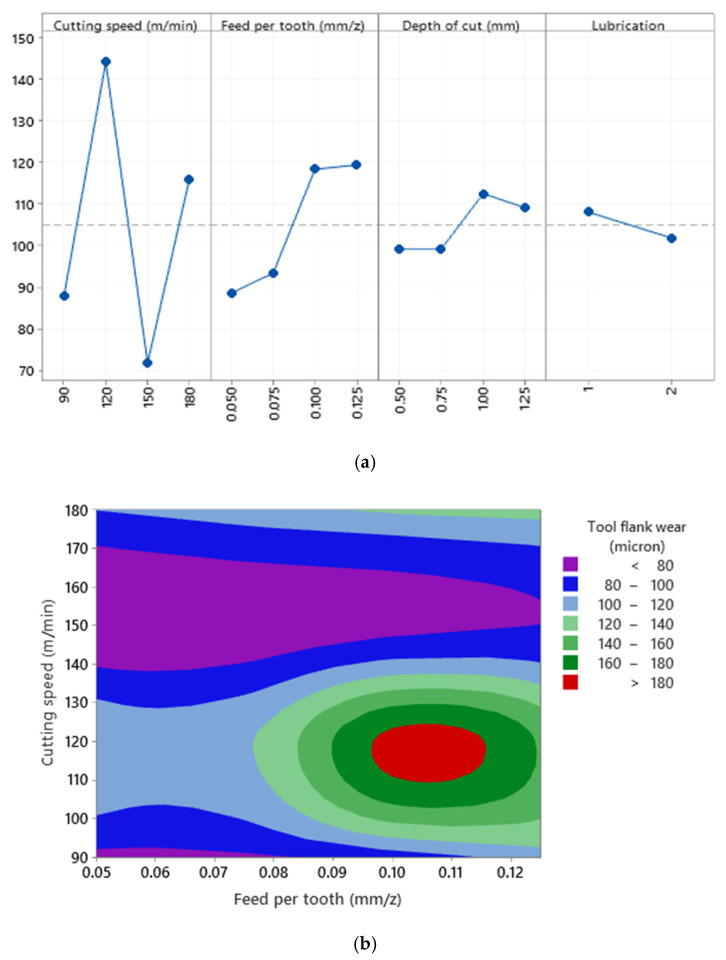
(**a**) The main effects plot of tool flank wear when machining Al520 + 10% SiC (R^2^ = 94.77%), (**b**) surface plot of tool flank wear versus cutting speed and feed per tooth.

**Figure 14 materials-15-01533-f014:**
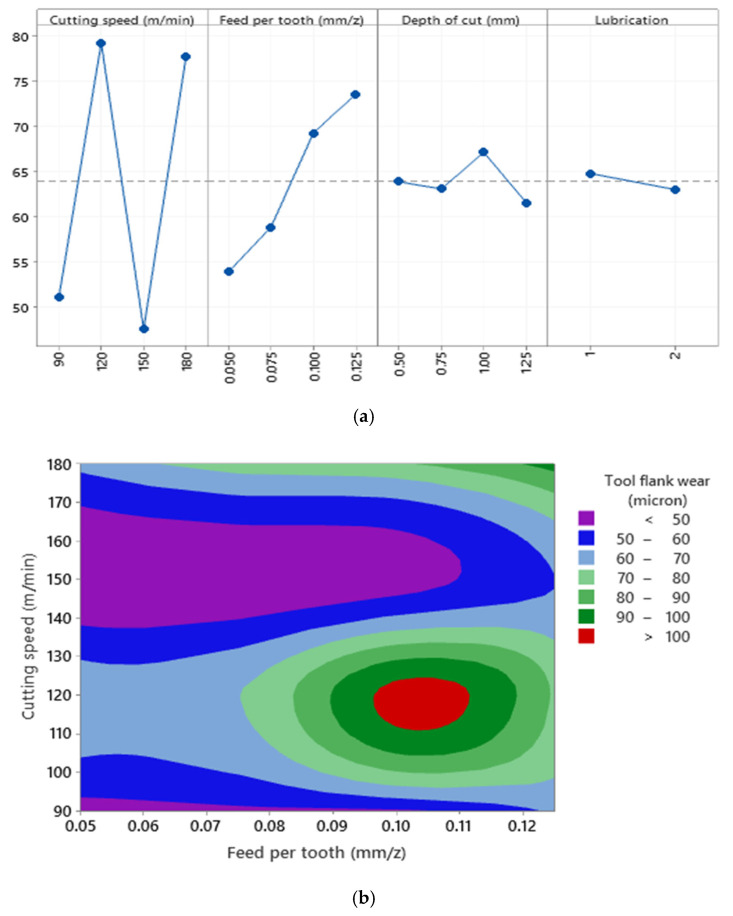
(**a**) The main effects plot of tool flank wear when machining Al520 + 10% SiC + 1% Bi; (R^2^ = 91.04%), (**b**) surface plot of tool flank wear versus cutting speed and feed per tooth.

**Figure 15 materials-15-01533-f015:**
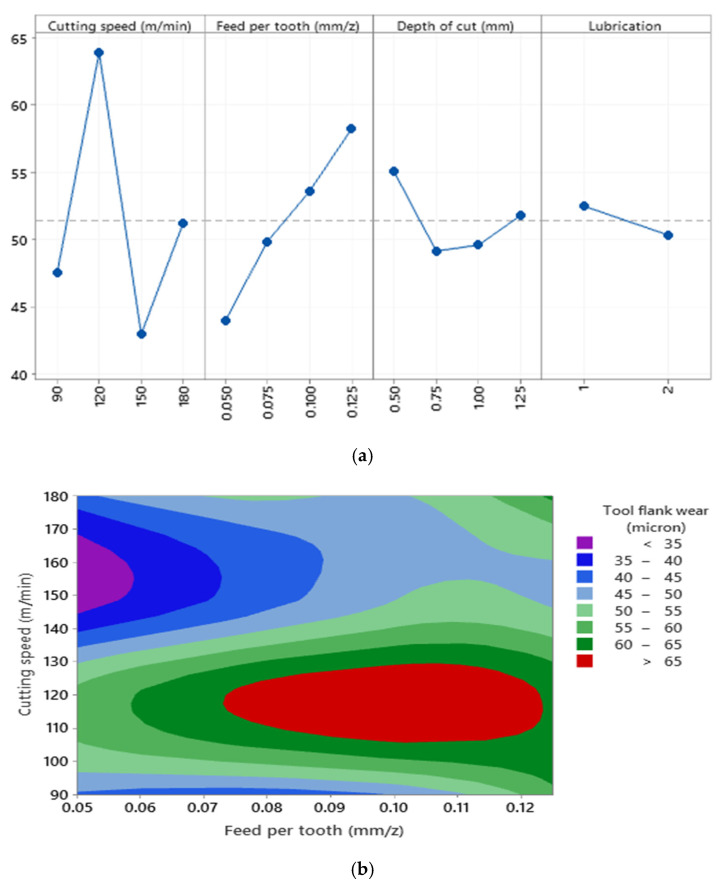
(**a**) The main effects plot of tool flank wear when machining Al520 + 10% SiC + 1% Sn; (R^2^
*=* 96.04%), (**b**) surface plot of tool flank wear versus cutting speed and feed per tooth.

**Table 1 materials-15-01533-t001:** The fabricated composite materials.

Composite Materials	Base Material	Reinforcement Particles
1	Al520	---
2	Al520	10% SiC
3	Al520	10% SiC + 1% Bi
4	Al520	10% SiC + 1% Sn

**Table 2 materials-15-01533-t002:** Quantometry results of Al-MMC.

Materials	Al	Si	Fe	Cu	Mn	Mg	Zn	Cr	Ti
Al520	89.32	0.083	0.0527	0.0225	0.0218	10.4	0.0025	0.0024	0.0025
Al520 + 10%SiC	84.11	4.44	0.133	0.0063	0.0239	9.7	0.0042	0.0115	0.0044
Al520 + 10%SiC + 1%Bi	89.06	0.21	0.238	0.005	0.0362	10.4	0.0194	0.0252	0.001
Al520 + 10%SiC + 1%Ni	88.47	2.36	0.0999	0.0675	0.0228	8.72	0.0052	0.0053	0.0042

**Table 3 materials-15-01533-t003:** Experimental parameters used.

Cutting Parameters	Levels			
Cutting speed (m/min)	90	120	150	180
Feed per tooth (mm/z)	0.05	0.075	0.1	0.125
Depth of cut (mm)	0.5	0.75	1	1.25
Cooling conditions	Dry	Wet	-	-

**Table 4 materials-15-01533-t004:** The DOE-L16 orthogonal array used for each tested MMC.

Test No.	Cutting Parameters			
	Lubrication Mode	Cutting Speed (m/min)	Feed Rate (mm/z)	Depth of Cut (mm)
1	Dry	90	0.05	0.5
2	Wet	90	0.075	0.75
3	Wet	90	0.1	1
4	Dry	90	0.125	1.25
5	Dry	120	0.05	0.75
6	Wet	120	0.075	0.5
7	Wet	120	0.125	1.25
8	Dry	120	0.1	1
9	Wet	150	0.05	1
10	Dry	150	0.75	1.25
11	Dry	150	0.1	0.5
12	Wet	150	0.125	0.75
13	Wet	180	0.05	1.25
14	Dry	180	0.75	1
15	Dry	180	0.1	0.75
16	Wet	180	0.125	0.5

**Table 5 materials-15-01533-t005:** Characteristics of the coated carbide insert used in end milling tests.

Grade	ap (mm)	Rɛ (mm)	W (mm)	F (mm)	S (mm)	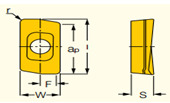
HP E90AN-D.50-2-C.50-7C CUTTER	7.7	0.4	4.5	1.2	2.6	PVD TiAlN coated carbide with W and Co as core and binder

**Table 6 materials-15-01533-t006:** The R^2^ of Ra, Rz in the tested materials within three measuring zones.

Materials	Measurement Sections					
	Enter		Middle		Exit	
	Ra	Rz	Ra	Rz	Ra	Rz
Al520	54.9	48.11	72.71	46.88	69.26	61.42
Al520 + 10% SiC	60.76	59.91	61.58	59.99	33.67	64.10
Al520 + 10%SiC + 1%Bi	37.17	63.40	63.25	68.67	52.89	87.72
Al520 + 10%SiC + 1%Sn	60.76	56.64	75.36	68.75	86.41	67.69

**Table 7 materials-15-01533-t007:** Order of importance of cutting parameters and conditions on tool flank wear when machining the tested materials.

Materials	Cutting Speed	Feed Rate	Depth of Cut	Lubrication Mode	R^2^
Al520	2	3	4	1	98.52%
Al520 + 10% SiC	1	2	4	3	94.77%
Al520 + 10% SiC + 1% Bi	1	2	4	3	91.04%
Al520 + 10% SiC + 1%Sn	1	2	4	3	96.04%

## Data Availability

Data sharing is not applicable.

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
