# Peer review of "Milling Al520-MMC Reinforced with SiC Particles and Additive Elements Bi and Sn"

_materials, 2022, doi:10.3390/ma15041533_

Round 1

Reviewer 1 Report

Some major revision should be done before further consideration.

1- In the title, Sic should be written as SiC.

2- Please be consistent in the font size in whole manuscript.

3- The abstract is too long and should be summarized. 

4- Most of the references in the introduction section are out of date and should be updated. The authors are recommended to read and use the following references in order to improve the quality of their introduction section: Journal of Composite Materials, 2015, 49 (28), 3507-3514; Journal of Composite Materials, 2018, 52 (27), 3745-3758; 

5- Too many figures in the manuscript, for examples figure 7-18 can be presented in 2 or 3 images.

6- Fig. 2 should be removed.

7- Figs. 1 and 3 should be merged.

8- Schematic of cutting process should be presented instead of real photograph of the process in Fig. 4.

9- Fig. 5, the SEM images are not clear. Please enlarge them and redraw the scale bar for more visibility in all element mappings.

Author Response

Dear Editor,

We would like to thank you for considering our paper entitled “Milling Al520-MMC reinforced with SiC particles and additive elements Bi and Sn” for possible publication in The International Journal of Advanced Manufacturing Technology. We are also very grateful to the respectful reviewers for their careful reviews and valuable suggestions. In the revised manuscript, all editorial comments have been fully incorporated. The following explanations are presented to these comments.

We look forward to hearing your constructive comments.   

Sincerely yours,

Seyed Ali Niknam, Ph.D

Reviewer 2 Report

The paper presents experimental aspects regarding the milling process of aluminum (Al520) - metal matrix composites reinforced with Sic particles and additive elements Bi and Sn. It is investigated the effects of cutting parameters and reinforcing elements on the surface roughness (Ra and Rz parameters) and flank wear.

In the article, the following points have to be improved: 

  • Rewrite the article according to the requirements of the journal; use the following general structure: Introduction, Materials and Methods, Results, Discussion and Conclusions;
  • Do not use abbreviations in the abstract;
  • In the Introduction section are necessary to be presented recent studies and research (from last years) related to the article topic;
  • Write the units in tables 4 and figures 7,9,11,13,15-18;
  • Use, in section 3, complete titles for the subsections, not only abbreviations names (Al 520, Al 520 + 10% Sic, Al 520 + 10%Sic + 1%Bi etc);
  • Rewrite the Conclusion section to emphasize what is new in the research and the presentation must be not like a numbered list of paragraphs;
  • Add new references published in the last 5 years. In the paper are not identified such references.

Author Response

(The authors gave the same response as above.)

Reviewer 3 Report

Experimental and statistical studies were carried out in the `Milling Al520-MMC reinforced with Sic particles and additive elements Bi and Sn` study. I am stating some formal corrections regarding the work done below. In particular, the evaluation of the results obtained should be revised and supported by the literature.
In the abstract part, it is not correct to say that there are no studies on this subject and that no results were found. These expressions should be simplified further.
In many places, the expression SiC is written as Sic. It should be corrected as SiC.
Line 102-s. `The latter operation eliminates moisture in the particles and improves their wetting properties.`
According to what this parameter is selected, it should be supported by the literature.

Line 109 – should be `and 5 grams of Bi and Sn were`
The writing of the figures should be in accordance with the magazine format.
Table.5 should be corrected.
In Figure 5, the resolution of the SEM pictures is very poor, and the dimensions at the bottom of the SEM pictures should also be shown.
In the result and discussion section, the titles should be rearranged and the intermediate titles in the form of 3-3.1-3.11 should be renamed. It should also be `4.Conclusion`.
Besides some of the figural errors I have mentioned, the evaluation of the results is very inadequate. The results obtained should be taken together with the literature. Result and Conclusion parts should be reinterpreted as a whole. Many studies have been carried out, but the evaluation of the obtained results is insufficient.

Author Response

(The authors gave the same response as above.)

Round 2

Reviewer 2 Report

The paper could be accepted as it is.

Reviewer 3 Report

The corrections are enough. The article is acceptable as it is.